# Real-time scene classification of unmanned aerial vehicles remote sensing image based on Modified GhostNet

**Xiaole Shen, Hongfeng Wang, Biyun Wei, Jinzhou Cao** *

College of Big Data and Internet, Shenzhen Technology University, Shenzhen, China

* caojinzhou2011@gmail.com

**Data Availability Statement:** All Files are available from the NWPU-RESISC database (url(s) https://hyper.ai/datasets/5449), UCMerced database (url(s) https://hyper.ai/datasets/5431), and AID database (url(s) https://hyper.ai/datasets/5446).

## Abstract

Unmanned Aerial Vehicles (UAVs) play an important role in remote sensing image classification because they are capable of autonomously monitoring specific areas and analyzing images. The embedded platform and deep learning are used to classify UAV images in real-time. However, given the limited memory and computational resources, deploying deep learning networks on embedded devices and real-time analysis of ground scenes still has challenges in actual applications. To balance computational cost and classification accuracy, a novel lightweight network based on the original GhostNet is presented. The computational cost of this network is reduced by changing the number of convolutional layers. Meanwhile, the fully connected layer at the end is replaced with the fully convolutional layer. To evaluate the performance of the Modified GhostNet in remote sensing scene classification, experiments are performed on three public datasets: UCMerced, AID, and NWPU-RESISC. Compared with the basic GhostNet, the Floating Point Operations (FLOPs) are reduced from 7.85 MFLOPs to 2.58 MFLOPs, the memory is reduced from 16.40 MB to 5.70 MB, and the predicted time is improved by 18.86%. Our modified GhostNet also increases the average accuracy (Acc) (4.70% in AID experiments, 3.39% in UCMerced experiments). These results indicate that our Modified GhostNet can improve the performance of lightweight networks for scene classification and effectively enable real-time monitoring of ground scenes.

## Introduction

Unmanned Aerial Vehicles (UAVs) as a remote sensing platform are used in various applications, such as search and rescue [1], disaster evaluation, and traffic monitoring [2], wireless communications [3]. Deep learning networks and embedded devices are equipped with UAVs to autonomously carry out tasks of image classification. An autonomous UAV rapidly monitors hazards and disasters by classifying the captured images in real-time, which relies heavily on its onboard sensors and microprocessors [4]. Local embedded devices are superior to cloud storage in scenarios involving privacy, latency, and limited connectivity. However, because embedded devices have limitations in memory and computing power, there are challenges to efficient scene classification.

**Funding:** Funding: This research is supported by National Natural Science Foundation of China (Grant No. 42001393, 41501370 and 62176165), the 5th College-enterprise Cooperation Project of Shenzhen Technology University (Grant No. 2021010802014), Shenzhen Science and Technology Program (Grant No. JCYJ20220530152817039), and Guangdong Science and Technology Strategic Innovation Fund (the Guangdong–Hong Kong-Macau Joint Laboratory Program, Grant No. 2020B1212030009). The funders had no role in study design, data collection and analysis, decision to publish, or preparation of the manuscript.

**Competing interests:** The authors have declared that no competing interests exist.

Scene classification is an important problem for remote sensing image classification. In recent years, numerous methods have been applied to complete the task of remote sensing scene classification. These methods can be divided into three categories [5]: handcrafted feature-based methods [6–10], unsupervised-feature-learning-based methods [11–17], and deep-learning-based methods. Handcrafted feature-based methods include SIFT (scale-invariant feature transform) [18], SURF(speeded-up robust features), HOG (histogram of oriented gradients) [8], etc. The unsupervised feature-learning-based methods automatically learn features from unlabeled images. K-Means [19] is a typical method. Compared with these two methods, deep learning has two characteristics of feature learning and deep structure, which is beneficial to achieving higher classification accuracy [20]. Most deep learning methods are based on Convolutional Neural Networks (CNNs) [21–24]. By deepening the number of network layers, CNNs can improve classification accuracy, such as AlexNet [25], Network in Network (NIN) [26], VGGNet [22], deep residual networks (ResNet) [23], etc.

However, the inference speed of deep learning networks on embedded platforms is impeded by the high computational cost, which is prohibitive from the perspective of memory consumption and real-time performance [27, 28]. The large memory footprint and high latency of deep learning networks are unacceptable for embedded devices in some special situations [29, 30].

Lightweight networks have been extensively studied to address these problems. Iandola et al. [31] first proposed the lightweight network SqueezeNet, whose parameter size can be reduced to less than 5MB, and the network parameter size can be further reduced by network compression. However, the SqueezeNet cannot be used in embedded devices, due to the lack of real-time performance. In 2017, a lightweight network MobileNet V1 [32] specifically for mobile devices, was proposed by the Google team. The Face++ team learned from the idea of MobileNet V1 and proposed ShuffleNet [33]. ShuffleNet used the method of shuffling channels to solve the problem of "sluggish information flow" caused by depthwise separable convolution. Experiments indicated that its classification accuracy is higher than that of MobileNet V1. In subsequent research, MobileNet V2 [34], ShuffleNet V2 [35] and MobileNet V3 [36] were proposed successively. In 2020, Huawei Noah Lab, together with Peking University and the University of Sydney, jointly launched a lightweight network model called GhostNet [37] that uses linear operations instead of convolution operations to analyze the phenomenon of redundancy in feature images. GhostNet is widely used in image classification, semantic segmentation, instance segmentation, target detection, etc.

In this paper, we build a Modified GhostNet based on the original GhostNet that can be deployed on an embedded device for real-time classification of drone imagery scenes. Data augmentation methods are employed to expand the diversity of samples in the UAV image scene dataset, including geometric transformation, pixel color transformation, and compound transformation. These data augmentation methods are verified on three datasets, including UCMerced, AID, and NWPU-RESISC. The Modified GhostNet model is trained based on transfer learning. The weights obtained by training on NWPU-RESISC are used as pre-training weight parameters, and weight transfer training is used for UCMerced and AID. The performance of the Modified GhostNet is evaluated in terms of FLOPs, memory usage, predicted time, and Acc. Compared with MobileNetV3-Small and the original GhostNet, the Modified GhostNet proposed in this paper reduces the amount of computation, thus improving real-time processing rate while reducing memory usage.

## Materials and methods

The contribution of this research is the development of a lightweight network based on the GhostNet model using visualization operation and structural compression techniques. We

analyze the basic architecture of the GhostNet model and then adjust the structure of the network, developing a Modified GhostNet model that has less network complexity while ensuring the accuracy of the model. The Modified GhostNet model designed in this paper can be deployed on an embedded device for image scenes classification with high requirements for both accuracy and real-time processing.

## GhostNet model

The GhostNet model is a lightweight network model jointly launched by Huawei Noah Lab, Peking University, and the University of Sydney in 2020. The model determined that during the training process of the network, there will be feature redundancy in each output layer in the middle. The conclusion is that a linear operation with a lower computational cost can replace a part of the convolution operation with a higher computational cost to reduce the amount of computation and save computing resources. The author proposes the Ghost module, which is characterized by replacing half of the convolution operations with linear operations, to reduce the computational complexity of the model. The structure of original GhostNet model is shown in Fig 1, including ordinary convolutional layers and Ghost module.

The compression ratio between the network model after using the Ghost module and the network model with pure convolution operation can reach 2 times. Suppose $n$ is the number of feature maps that a convolution layer should produce, $m$ represents the actual number of feature maps output by the convolution operation, and $s$ represents the factor of the linear operation. Since the general convolution operation is replaced by a linear operation in this paper, $s = 2$. Then, according to the principle that the total number of feature maps output by a certain layer should remain unchanged, we know that $n = m \cdot s$, which is equivalent to $m \cdot (s - 1) = \frac{n}{s} \cdot (s - 1)$. Suppose the average size of the convolution kernel for each linear operation is $d \times d$. The compression ratio formula between the network model after using the Ghost module and the network model with pure convolution operation can be obtained by

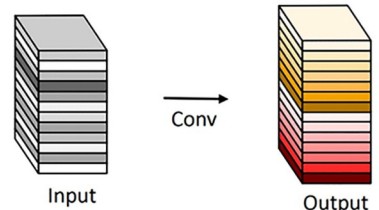

(a) The convolutional layer.

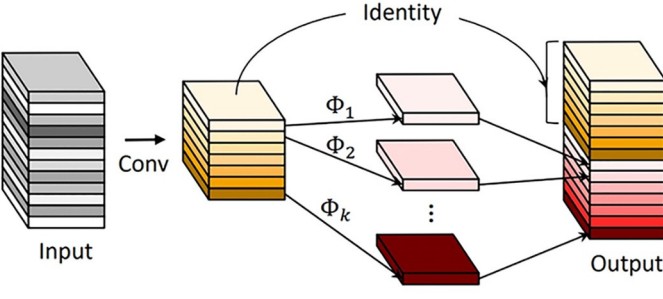

(b) The Ghost module.

**Fig 1. Structure of original GhostNet model.** a. The convolutional layer, b. The Ghost module.

calculation, which is shown below, where $c$ is the number of input channels, and $k$ is the size of the convolution kernel of the convolution operation.

$$rc = \frac{n \cdot c \cdot k \cdot k}{\frac{n}{s} \cdot c \cdot k \cdot k + \frac{s-1}{s} \cdot d \cdot d} \approx \frac{s \cdot c}{s + c - 1} \approx s \tag{1}$$

The author designed two residual structures that make up the Ghost bottlenecks of Ghost-Net model, and two structures with a stride of 1 and a stride of 2. The two structures are shown in Fig 2. Ghost bottleneck is mainly composed of two Ghost modules stacked. For stride of 2, the shortcut path uses the downsampling layer and inserts depthwise convolution with the stride of 2 in the middle of the Ghost module. The original GhostNet model is formed by stacking the bottlenecks of the two sub-network models. The overall architecture is shown in Table 1.

### Modified GhostNet model

In this paper, we propose a network model based on structural compression. The feature map redundancy phenomenon is analyzed when the GhostNet model is applied to the UAV image scene classification dataset. We remove the Ghost bottlenecks that generate redundant feature maps in the GhostNet model and replace the final fully connected layer with a fully convolutional layer. The Modified GhostNet has lower memory usage and energy consumption. It is well known that computationally intensive models drain batteries quickly for embedded devices. The same embedded device processes a model with a large amount of calculation. If the battery can only support 30 minutes, then reduce the amount of calculation by half, and it can work for at least 10 more minutes. Therefore, reducing the computational load of network models is crucial for supporting models in embedded devices. The improvement scheme of the GhostNet model is mainly based on the following two points:

1. The 2nd, 4th, 9th, 10th, 15th, and 17th layers in the GhostNet model are extracted. Through the visualization operation, it is found that there is a large amount of feature redundancy between these layers and their adjacent layers;

2. The fully connected layer at the end is replaced with a fully convolutional layer to reduce the amount of network computation caused by the fully connected layer.

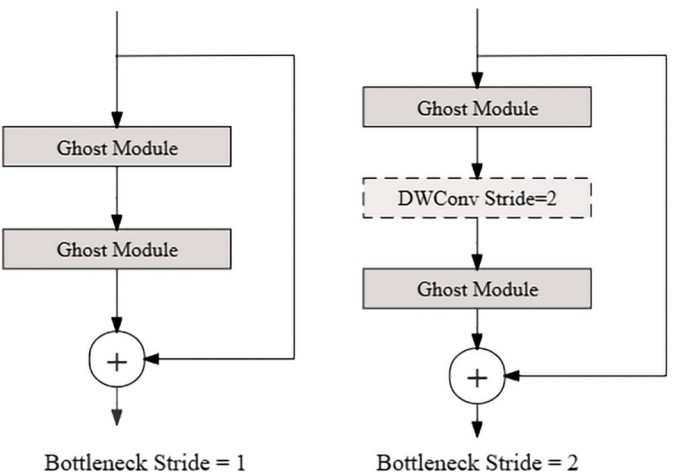

**Fig 2. Structure of Ghost bottlenecks.**

**Table 1. Original overall architecture of GhostNet.**

| Input | Operator | Expansion channel | Output channel | SE | Stride |
|---|---|---|---|---|---|
| 256*256*3 | Conv2d 3*3 | - | 16 | - | 1 |
| 112*112*16 | GBneck | 16 | 16 | 1 | 2 |
| 56*56*16 | GBneck | 4 | 24 | - | 1 |
| 28*28*24 | GBneck | 72 | 24 | - | 1 |
| 28*28*24 | GBneck | 72 | 40 | 1 | 2 |
| 14*14*40 | GBneck | 120 | 40 | 1 | 1 |
| 14*14*40 | GBneck | 240 | 80 | - | 2 |
| 14*14*80 | GBneck | 200 | 80 | - | 1 |
| 14*14*80 | GBneck | 184 | 80 | - | 1 |
| 14*14*80 | GBneck | 184 | 80 | - | 2 |
| 14*14*80 | GBneck | 480 | 112 | 1 | 1 |
| 7*7*160 | GBneck | 960 | 160 | - | 1 |
| 7*7*160 | GBneck | 960 | 160 | 1 | 1 |
| 7*7*160 | GBneck | 960 | 160 | - | 1 |
| 7*7*160 | GBneck | 960 | 160 | 1 | 1 |
| 7*7*160 | Conv2d 1*1 | - | 960 | - | 1 |
| 7*7*960 | GAP 7*7 | - | - | - | - |
| 1*1*960 | Conv2d 1*1 | - | 1280 | - | 1 |
| 1*1*1280 | FC | - | 1000 | - | - |

The structure of Modified GhostNet network model based on these two points is shown in Table 2. K is the number of classification samples, and SE indicates whether to use the Squeeze-And-Excite module.

## Experiment

### Dataset

With the development of neural networks, the number of data samples, the amount of computation, and the algorithm have become the three key factors that have the greatest impact on

**Table 2. Modified overall architecture of GhostNet.**

| Input | Operator | Expansion channel | Output channel | SE | Stride |
|---|---|---|---|---|---|
| 256*256*3 | Conv2d 3*3 | - | 16 | - | 2 |
| 112*112*16 | GBneck 3*3 | 16 | 16 | 1 | 2 |
| 5the 6*56*16 | GBneck 3*3 | 72 | 24 | - | 2 |
| 28*28*24 | GBneck 5*5 | 88 | 24 | - | 1 |
| 28*28*24 | GBneck 5*5 | 96 | 40 | 1 | 2 |
| 14*14*40 | GBneck 5*5 | 240 | 40 | 1 | 1 |
| 14*14*40 | GBneck 5*5 | 120 | 48 | 1 | 1 |
| 14*14*48 | GBneck 5*5 | 144 | 48 | 1 | 1 |
| 14*14*48 | GBneck 5*5 | 288 | 96 | 1 | 2 |
| 7*7*96 | GBneck 5*5 | 576 | 96 | 1 | 1 |
| 7*7*96 | GBneck 5*5 | 576 | 96 | 1 | 1 |
| 7*7*96 | Conv2d 1*1 | - | 576 | - | 1 |
| 7*7*576 | GAP 7*7 | - | - | - | - |
| 1*1*576 | Conv2d 1*1 | - | 1024 | - | 1 |
| 1*1*1024 | Conv2d 1*1 | - | K | - | - |

**Table 3. The parameters of UAV images dataset.**

| Dataset | Samples | Categories | Number of per category | Image size | Spatial resolution |
|---|---|---|---|---|---|
| UCMerced | 2100 | 21 | 100 | $200 \times 200$ | 0.3m |
| AID | 10,000 | 30 | 200-400 | $600 \times 600$ | 0.5-0.8m |
| NWPU-RESISC | 31,500 | 45 | 700 | $256 \times 256$ | 2-30m |

the neural network model. Among them, sample data is the basis for all research work. When the dataset used in the research is not suitable, that is, the number of samples in the dataset is scarce or the data samples do not have diversity, it will be difficult to carry out the research work smoothly [38]. Brill et al. [39] gave relevant conclusions on this problem. The same problem has almost the same performance on different algorithms. If the sample data is increased, the overall accuracy of the algorithm will be improved to a certain extent.

The quality of the dataset is reflected in the richness and diversity of data samples. For natural images, the same type of samples should have different shapes, angles, sizes, etc. For UAV images, the same type of samples should include different factors such as climate change, illumination change, viewing angle change, and spatial resolution change. The current public UAV image scene datasets have some deficiencies. The commonly used public UAV image scene datasets are shared in Table 3, including UCMerced, AID and NWPU-RESISC [40].

UCMerced is a dataset with 21 categories that was released by the Computer Vision Laboratory of the University of California in 2010, which is the earliest UAV image dataset collected. However, the number of various types of samples in this dataset is relatively small, and the number of pictures in each category is only 100. Additionally, the scene information involved in this dataset is limited to American cities, and the diversity of samples is lacking.

Wuhan University and Huazhong University of Science and Technology released a UAV image dataset called AID. The dataset contains 30 categories of data, a total of 10,000 UAV images, and the resolution of each image is $600 \times 600$. There are about 220-420 pictures. Although the dataset has some variation in spatial resolution, the number of images per category is inconsistent.

NWPU-RESISC is a large-scale benchmark dataset created by Northwestern Polytechnical University in 2016, and it is the most commonly used dataset for validating network performance. Although there are certain advantages compared to the first two datasets, the number of types of samples in the dataset is relatively small, and the image data of similar samples are basically ideal environments and lack sample diversity.

## Data preprocessing

The dataset used in this paper has a small capacity, and the similarity of the data samples is extremely high, which may lead to an overfitting problem in the network model. Therefore, we utilize image augmentation methods to expand the capacity of image scene data. The image augmentation methods are used to enrich the scene information in different environments and improve the generalization capacity of the network. The approaches include geometric transformation, pixel color transformation, and complex transformation. The geometric transformation includes flipping, rotating, translating, scaling, cropping, and other operations. Pixel color transformation includes noise interference and blurring. Combination of different rotation angles and different noise is shown in Fig 3. In addition, the composite transformation is important for enriching the sample characteristics of UAV image datasets. Compound transformation is the combination of geometric transformation and pixel color transformation. Combining rotation and noise interference can simulate complex sample data taken from different directions. The

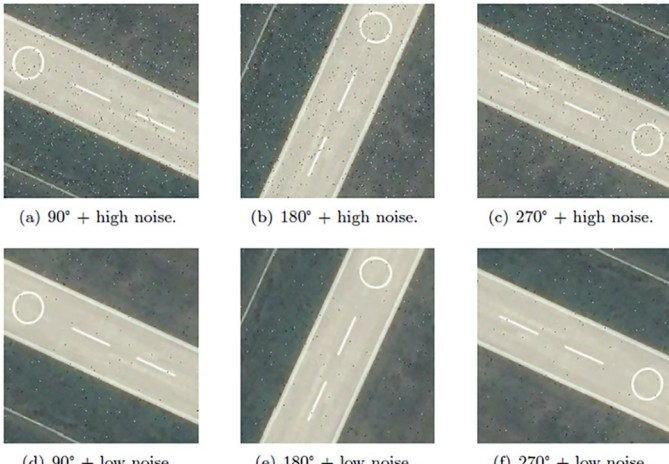

**Fig 3. Combination of rotation and noise interference.** a) 90˚ + high noise, b) 180˚ + high noise, c) 270˚ + high noise, d) 90˚ + low noise, e) 180˚ + low noise, f) 270˚ + low noise.

combination of rotation, brightness, and contrast augmentation can simulate pictures in different lighting conditions and in different shooting orientations. The combination of different rotation angles and different brightness and contrast augmentation is shown in Fig 4.

We use transfer learning techniques to generate the Modified GhostNet model. Transfer learning means that knowledge learned in one domain (such as knowledge learned in natural scenes) is applied to another domain (such as defect detection or drone images) to improve its generalization ability [41]. The pre-training model in this paper is the model trained on NWPU-RESISC. The weights generated in the model are set as the initial weights of AID and UCMerced, and then the network model is fine-tuned.

The neural network model must have feature redundancy during the training process. This paper uses the GhostNet network model to train using the NWPU-RESISC dataset and visualizes the feature maps of the output channels from the 8th to 11th layers and observes the

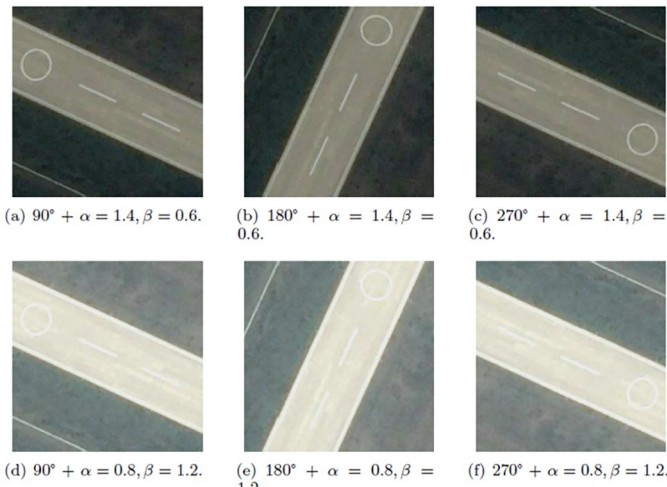

**Fig 4. Combination of rotation and different brightness and contrast augmentation.** (a) 90˚ + $\alpha = 1.4, \beta = 0.6$. (b) 180˚ + $\alpha = 1.4, \beta = 0.6$. (c) 270˚ + $\alpha = 1.4, \beta = 0.6$. (d) 90˚ + $\alpha = 0.8, \beta = 1.2$. (e) 180˚ + $\alpha = 0.8, \beta = 1.2$. (f) 270˚ + $\alpha = 0.8, \beta = 1.2$.

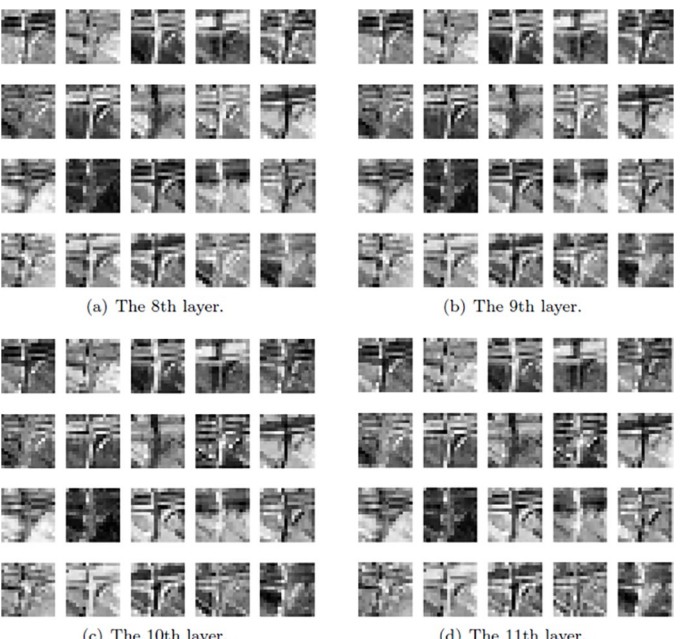

**Fig 5. Feature maps derived from different layers.** (a) The 8th layer. (b) The 9th layer. (c) The 10th layer. (d) The 11th layer.

continuous output layer. There is redundancy of feature maps. Here, 20% of the pictures are randomly selected for comparison, as shown in Fig 5. Feature redundancy is revealed in the training process for the UAV image dataset while using original GhostNet model, which is improved by Modified GhostNet.

## Experiment environment

To ensure the training speed of the model, the model training system environment used in this experiment is Ubuntu16.04, and the hardware platform is Quadro P5000. The model was built on the Keras deep learning framework. During the training process, the Adam is selected as optimizer, the initial learning rate is set to 0.001, the training rounds are set to 100, and the batch size is set to 8. The image resolution in the dataset is set to $256 \times 256$, and the same image augmentation operations are performed, including geometric transformation, pixel color transformation, and composite transformation.

Since the purpose of this project is to apply the lightweight network model to embedded devices, the trained network model is transplanted to the embedded device for prediction. The embedded device selected in this experiment is the Jetson TX2. The test method in this paper mainly adopts the simulation test. The embedded device is connected to a camera through USB, and the camera is treated to recognize the picture, and the recognized category is drawn in the upper left corner of the picture in the form of characters. The display screen is used to display the ground object information captured by the drone. On the left side of the Jetson TX2 device is a monocular camera, which is used to capture the ground object information.

## Training strategy

To ensure the training speed of the model, the model training system environment used in this experiment is Ubuntu16.04, and the hardware platform is Quadro P5000. The building of the model was done on the Keras deep learning framework.

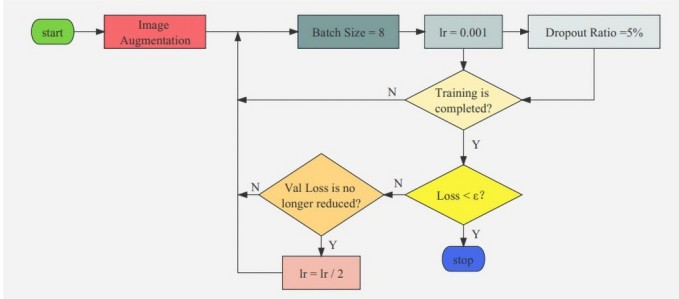

**Fig 6. The entire training strategy of the Modified GhostNet model.**

In this paper, the capacity of the dataset is expanded through the methods of image augmentation, which prevents overfitting in network training and improves the generalization ability of the network. The image resolution in the dataset is set to 256 × 256. The dataset is shuffled and divided into three parts in a ratio of 6:2:2, including training set, validation set and test set. During the training process, the Adam optimizer is used, the initial learning rate (lr) is set to 0.001, the training rounds are set to 100, the batch size is set to 8, and dropout ratio is set to 5%. The entire training strategy of Modified GhostNet model is shown in Fig 6. Val Loss is the loss of validation set. Training complete means training rounds reach 100 and $\varepsilon$ is a constant value.

## Results

### Image augmentation

To verify the generality of the scheme, this paper will conduct experiments on three small UAV image scene classification datasets: UCMercedx, AID, and UCMerced. Each sample data is expanded by 20 times the original, thus the total number of samples in the dataset is also 20 times the original. The division ratio of the training set, validation set, and test set of the dataset is 6:2:2. The number of a single category in training set before and after augmentation is shown in Table 4. The number of training samples is greatly increased, providing rich sample features for model training.

The dataset after image augmentation is employed trained on GhostNet. The average accuracy of GhostNet using dataset after image augmentation is improved, which is revealed in Table 5.

**Table 4. The samples of a single category in training set before and after augmentation.**

| Dataset | Samples (before) | Samples (after) |
|---|---|---|
| UCMerced | 60 | 1200 |
| AID | 120 | 2400 |
| NWPU-RESISC | 420 | 8400 |

**Table 5. The average accuracy of GhostNet before and after image augmentation.**

| Dataset | Acc. (before) | Acc. (after) |
|---|---|---|
| UCMerced | 79.60% | 92.80% |
| AID | 85.25% | 87.35% |
| NWPU-RESISC | 88.36% | 91.45% |

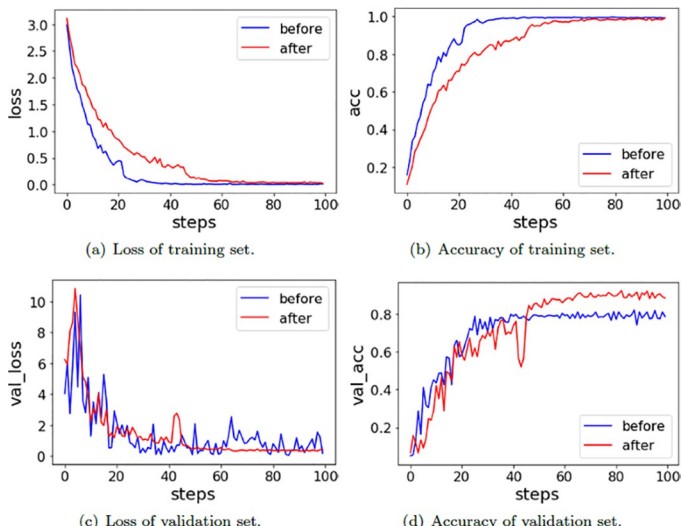

**Fig 7. The loss and accuracy of the GhostNet model before and after UCMerced image augmentation.** (a) Loss of training set. (b) Accuracy of training set. (c) Loss of validation set. (d) Accuracy of validation set.

**UCMerced.**   The loss and accuracy of the GhostNet model before and after UCMerced image augmentation are shown in Fig 7. The red line represents the change trend after image augmentation, the blue line represents the change trend before image augmentation, and steps represent the training rounds.

In Fig 7, figure(a) indicates that the convergence rate of the network model on UCMerced after image augmentation is slower than that before image augmentation. Figure(c) shows that the convergence of the network model before image augmentation on the validation set is always jittered within a certain range, and the convergence of the network model after image augmentation on the validation set maintains a relatively stable trend after 50 rounds of training. In comparing figure(b) and figure(d), it can be seen that the network model before image augmentation has a very serious overfitting phenomenon. The accuracy of the blue line is equal to the red line in the training set, and its accuracy is close to 100%. However, the accuracy of the blue line in the validation set is less than 80%, and the red line is not much different from the red line in the training set. The experimental results indicate that the preprocessing method of image augmentation is able to alleviate the overfitting phenomenon well.

**AID.**   Before experimenting with AID, this paper reduces the sample resolution in AID to $256 \times 256$, which is consistent with the resolution of the other two datasets. Fig 8 shows the loss and accuracy of the GhostNet model before and after AID image augmentation.

As shown in Fig 8, compared with the decrease in the loss of the training set before image augmentation, the value loss after image augmentation is lower, indicating that its convergence occurs before image augmentation. In figure(a), the loss before data enhancement is always in a jitter state, which is caused by the imbalance of sample data in the dataset before data enhancement. As shown in figure(b) and figure(d), the AID fitting situation before and after data enhancement is similar, but the overall accuracy after data enhancement is higher.

**NWPU-RESISC.**   Fig 9 shows the loss and accuracy of the GhostNet model before and after NWPU-RESISC image augmentation. The red line represents the change trend after data enhancement, the blue line represents the change trend before data enhancement, and steps represent the training rounds.

From figure(a) and figure(c), it can be seen that the loss of NWPU-RESISC after image augmentation is lower on the training set and validation set after 70 rounds of training. Figure(b)

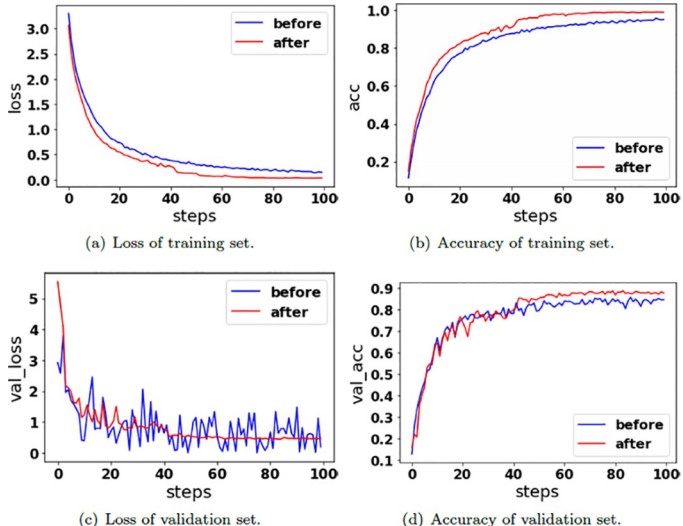

**Fig 8. The loss and accuracy of the GhostNet model before and after AID image augmentation.** (a) Loss of training set. (b) Accuracy of training set. (c) Loss of validation set. (d) Accuracy of validation set.

and figure(d) indicate that the accuracy of the validation set and training set has a small degree of overfitting before and after data enhancement, and the overall accuracy after data enhancement is higher than before image augmentation. Experimental results reveal that image augmentation can, indeed, improve the accuracy of the network to a certain extent.

## Comparison of models

To verify the validity of the GhostNet model based on structure compression, we compare MobileNetV3-Small, GhostNet, and the Modified GhostNet models through different evaluation metrics, including FLOPs, memory usage, predicted time, and average accuracy.

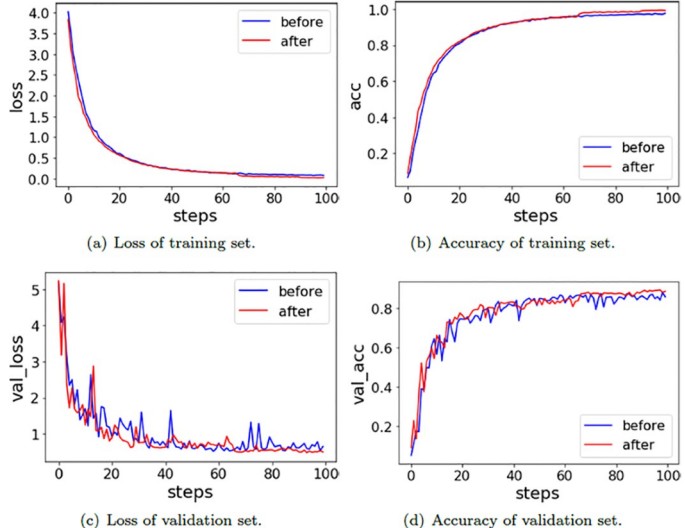

**Fig 9. The loss and accuracy of the GhostNet model before and after NWPU-RESISC image augmentation.** (a) Loss of training set. (b) Accuracy of training set. (c) Loss of validation set. (d) Accuracy of validation set.

**Table 6. FLOPs of different models on UCMerced dataset.**

| Model | FLOPs (M) |
|---|---|
| MobileNetV3-Small | 6.09 |
| GhostNet | 7.85 |
| Modified GhostNet | 2.58 |

**FLOPs.**   In lightweight neural networks, Floating Point Operations (FLOPs) are commonly used to measure the complexity of the network model. FLOPs represent floating point operands, also known as computations. The calculation formula of FLOPs is:

$$FLOPs_{CNN} = H \times W \times C_{in} \times C_{out} \times F^2 \qquad (2)$$

Among them, $H$ and $W$ represent the height and width of the input feature map, respectively, $C_{in}$ represents the number of channels of the input feature map, $C_{out}$ represents the number of channels of the output feature map, and $F$ represents the size of the convolution kernel.

Since the amount of computation generated in the model has nothing to do with the dataset and the structure of the model itself, only the amount of data on one dataset is shown below, and the others are the same. Table 6 shows the FLOPs of different models on UCMerced dataset. The amount of computation of the Modified GhostNet model is reduced by as much as 3 times that of the original GhostNet model. Compared with MobileNetV3-Small, it is also reduced by nearly 3 times.

**Memory usage and predicted time.**   Embedded devices have high requirements on model memory usage and real-time performance. The reduction of the parameter quantity is accompanied by the reduction of the memory occupancy rate, and the reduction of the calculation quantity is accompanied by the reduction of the operation time. The combined effect of these two factors is important for the predicted speed of the network model. The memory usage and predicted time of different models are shown in Table 7.

The quantitative results show that the amount of the Modified GhostNet's parameters is reduced by nearly 3 times that of the original GhostNet model, and the overall prediction time is also reduced from 52.5ms to 42.6ms. The Modified GhostNet model also shows its overall performance advantage over the MobileNetV3-Small model, both in terms of memory usage and forward inference time.

**Acc.**   Except for FLOPS, memory usage, and real-time performance, average accuracy (Acc) of models deployed on embedded devices is one of the most critical metrics. Acc of different models on the three datasets is displayed in Table 8.

Dropout with the ratio of 0.05 and weight transfer are employed in the Modified GhostNet model proposed in this paper. The average accuracy of three models on three dataset indicate that the Modified GhostNet has higher accuracy. On the UCMerced dataset, the average accuracy of Modified GhostNet is 96.19%, which performs better than MobileNetV3-Small and original GhostNet. The average accuracy of Modified GhostNet is 92.05% on AID dataset,

**Table 7. The memory usage and predicted time of different models.**

| Model | Memory usage (MB) | Predicted time on Jetson TX2 (ms) | Predicted time on Quadro P5000 (ms) |
|---|---|---|---|
| MobileNetV3-Small | 12.7 | 47.7 | 18.2 |
| Original GhostNet | 16.4 | 52.5 | 19.1 |
| Modified GhostNet | 5.7 | 42.6 | 17 |

**Table 8. The average accuracy of different models.**

| Model | Dataset | Acc. (%) |
|---|---|---|
| MobileNetV3-Small | UCMerced | 92.38 |
| Original GhostNet | UCMerced | 92.80 |
| Modified GhostNet | UCMerced | **96.19** |
| MobileNetV3-Small | AID | 87.25 |
| Original GhostNet | AID | 87.35 |
| Modified GhostNet | AID | **92.05** |
| MobileNetV3-Small | NWPU-RESISC | 91.36 |
| Original GhostNet | NWPU-RESISC | 91.45 |
| Modified GhostNet | NWPU-RESISC | **91.73** |

which is much higher than MobileNetV3-Small and original GhostNet. The Modified Ghost-Net has a certain improvement in accuracy compared with the original network structure, and the UCMerced and AID datasets have increased by 3.39% and 4.70% respectively. However, the average accuracy of the three models is similar on NWPU-RESISC dataset, and Modified GhostNet increased by 0.28% compared with original GhostNet.

The effect of dropout position on average accuracy is shown in Fig 10. Experiments are performed on three datasets, and dropout operations are performed on the 5th, 7th, 9th, and 11th layer of the Modified GhostNet. The dropout operation can improve the classification accuracy of the model, and the dropout in different positions has different effects on the classification accuracy of the model. In UCMerced and AID, adding dropout to the 9th layer can greatly improve the accuracy of the model, while in NWPU-RESISC the introduction of dropout at the 7th layer was most effective.

For small datasets, fine-tuning the network model by weight transfer can usually achieve good results. Weight transfer is a technique commonly used in transfer learning, where a pre-trained model on a large dataset is used as a starting point for training a model on a smaller dataset. NWPU-RESISC dataset is a larger dataset, while UCMerced and AID datasets are smaller. In this experiment, the weights obtained by training on NWPU-RESISC are used as pre-training weight parameters, and weight transfer training is used for UCMerced and AID.

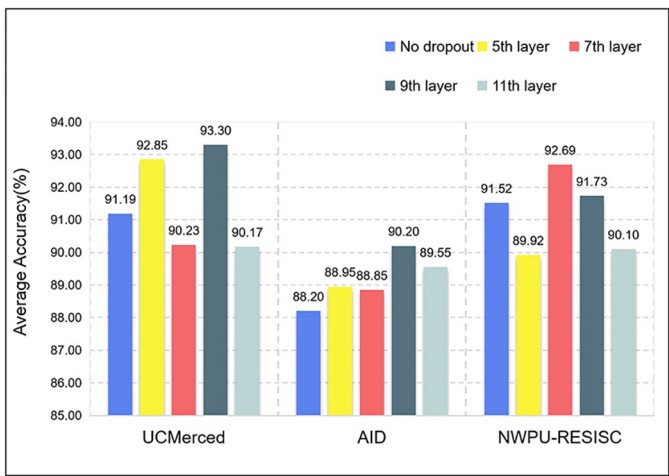

**Fig 10. The average accuracy of dropout position.**

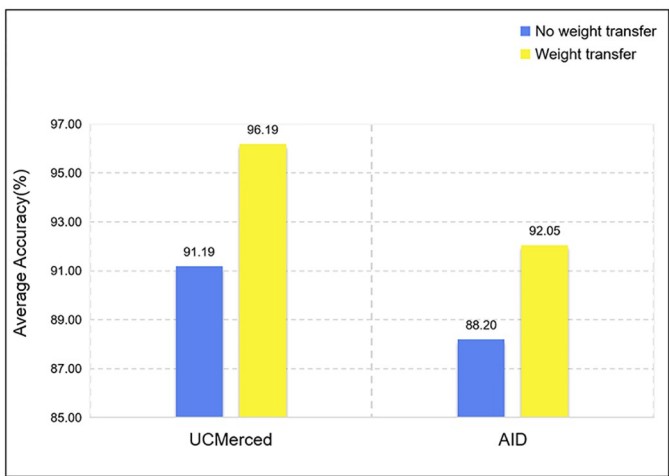

**Fig 11. The average accuracy of the model before and after using transfer learning.**

The policy tunes the network model to fit the respective dataset. As shown in Fig 11, the figure shows the change in the classification accuracy of the model before and after using transfer learning. The quantitative results show that adjusting the network model by weight transfer can avoid the phenomenon of network overfitting and effectively improve the prediction accuracy of the network.

## Discussion

The evaluation metrics examined in this study were FLOPs, memory usage, predicted time, and Acc. The values obtained from these criteria in the modified network are as follows. The FLOPs was 2.58 MFLOPs, the memory usage was 5.7 MB, the predicted time on Jetson TX2 was 42.6 ms, the Acc On the UCMerced dataset was 96.19, and the Acc on the AID dataset was 92.05. Based on the basic GhostNet results, the proposed network had an average improvement of 5.27 in FLOPs, 18.86% in predicted time, 3.39% in Acc on UCMerced dataset, and 4.70% in Acc on AID dataset, indicating an improvement in the classification of the proposed model.

In recent years, the employment of artificial intelligence and deep learning methods has become one of the most popular and useful approaches in scene classification. LeCun et al. [42] introduced the LeNet5 convolutional neural network model. Lin et al. [26] proposed Network in Network (NIN). The entire network model was formed by stacking sub-networks and could be changed arbitrarily. Based on the idea of increasing network depth, VGG [22] increased the depth of the CNN model to about 20 layers. The error rate of image recognition has also decreased from 11% to 6.7%, which is gradually approaching the human eye error rate of 5.1%. He et al [23] formulated ResNet, improving the network accuracy by increasing the network depth. However, the real-time performance of these studies still needs improvement. In addition, one of the limitations of these studies was that the computing performance of most embedded devices cannot support the deep convolutional neural network model.

Previous research has focused on improving the accuracy, with a few studies being actually applied to embedded environments. Moreover, no matter how high the classification accuracy is, the experimental conditions are tested on high-performance equipment in the laboratory, and the network model does not go out of the laboratory and into practice. Thus, it can be said that porting neural network models to embedded devices is still an important issue for UAV

image scene classification. The main goal of this study is to reduce computational costs and apply the lightweight network model to real-time scene classification of UAV images.

In this study, the GhostNet was modified to improve the challenges of distinguishing ground objects from UAV images in real-time. The Modified GhostNet model can not only reduce the memory and calculation amount of the model, thereby improving the prediction speed of the model, but also improve the accuracy of some data sets. Because the network model with high complexity easily causes overfitting when training on small data sets, the structure compression of the network model can reduce the complexity of the network and alleviate the overfitting phenomenon to some extent. Compared with MobileNetV3 and the basic GhostNet, the Modified GhostNet has faster speed and higher classification accuracy.

## Conclusion

Based on the original GhostNet, we built a lightweight neural network, the Modified GhostNet model, which can be transplanted on an embedded device for real-time UAV classification of drone imagery scenes. We utilize image augmentation methods to expand the sample diversity in the UAV image dataset, including geometric transformation, pixel color transformation, and compound transformation. The Modified GhostNet model is transplanted to the embedded device Jetson TX2, which is trained on three datasets based on transfer learning. The performance of the Modified GhostNet is evaluated in FLOPs, memory usage, predicted time, and Acc. Compared with MobileNetV3-Small and original GhostNet, the Modified GhostNet proposed in this paper reduces the amount of computation, improving real-time processing rate while reducing memory usage.

## Author Contributions

**Conceptualization:** Xiaole Shen, Jinzhou Cao.

**Data curation:** Biyun Wei.

**Formal analysis:** Hongfeng Wang.

**Funding acquisition:** Xiaole Shen, Jinzhou Cao.

**Investigation:** Biyun Wei.

**Methodology:** Xiaole Shen, Hongfeng Wang.

**Project administration:** Xiaole Shen, Jinzhou Cao.

**Resources:** Xiaole Shen.

**Software:** Hongfeng Wang, Biyun Wei.

**Supervision:** Xiaole Shen, Jinzhou Cao.

**Validation:** Hongfeng Wang, Biyun Wei.

**Visualization:** Hongfeng Wang, Biyun Wei.

**Writing – original draft:** Hongfeng Wang, Biyun Wei.

**Writing – review & editing:** Xiaole Shen, Hongfeng Wang, Jinzhou Cao.

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
