## [Decision Letter · Decision Letter 0]

24 Apr 2023

PONE-D-23-05971Real-time Scene Classification of UAV Remote Sensing Image based on Modified GhostNetPLOS ONE

Dear Dr. Cao,

Thank you for submitting your manuscript to PLOS ONE. After careful consideration, we feel that it has merit but does not fully meet PLOS ONE’s publication criteria as it currently stands. Therefore, we invite you to submit a revised version of the manuscript that addresses the points raised during the review process.

We look forward to receiving your revised manuscript.

Kind regards,

Sunday Adeola Ajagbe

Academic Editor

PLOS ONE

“Funding: This research is supported by “National Natural Science Foundation of China (NSFC)”(https://www.nsfc.gov.cn/) (Grant No.  41501370 and 62176165),  and “the 5th College-enterprise Cooperation Project of Shenzhen Technology University” (https://english.sztu.edu.cn/)(Grant No. 2021010802014).”

4. We note that Figure 3 in your submission contain [map/satellite] images which may be copyrighted. All PLOS content is published under the Creative Commons Attribution License (CC BY 4.0), which means that the manuscript, images, and Supporting Information files will be freely available online, and any third party is permitted to access, download, copy, distribute, and use these materials in any way, even commercially, with proper attribution. For these reasons, we cannot publish previously copyrighted maps or satellite images created using proprietary data, such as Google software (Google Maps, Street View, and Earth). For more information, see our copyright guidelines: http://journals.plos.org/plosone/s/licenses-and-copyright.

a. You may seek permission from the original copyright holder of Figure 3 to publish the content specifically under the CC BY 4.0 license. 

Additional Editor Comments:

The application of UAV can be enriched by description of the introduction section. The authors can refer to papers

Affum, E. A., Adigun, M. O., Boateng, K. A., Ajagbe, S. A., Addo, E. (2022). Enhancing UAV Communication Performance: Analysis Using Interference Based Geometry Stochastic Model and Successive Interference Cancellation. In: Gervasi, O., Murgante, B., Hendrix, E.M.T., Taniar, D., Apduhan, B.O. (eds) Computational Science and Its Applications – ICCSA 2022. ICCSA 2022. Lecture Notes in Computer Science, vol 13375. Springer, Cham. https://doi.org/10.1007/978-3-031-10522-7_17

Reviewers' comments:

Reviewer's Responses to Questions

**Comments to the Author**

1. Is the manuscript technically sound, and do the data support the conclusions?

Reviewer #1: Yes

Reviewer #2: Yes

2. Has the statistical analysis been performed appropriately and rigorously? 

Reviewer #1: Yes

Reviewer #2: Yes

3. Have the authors made all data underlying the findings in their manuscript fully available?

Reviewer #1: No

Reviewer #2: Yes

4. Is the manuscript presented in an intelligible fashion and written in standard English?

Reviewer #1: Yes

Reviewer #2: Yes

5. Review Comments to the Author

Reviewer #1: The authors should write UAV in full for the paper title.

Under the Abstract section, what does the authors mean by M as the measurement unit? Is it MB or Mb or M what?

Generally,

* Placing the Figures very far from where it was cited in the body of the paper makes it difficult to really see if the

Figures are in order for the purpose mentioned or not. E.g. There are 2 Figures (a & b) on the page referred to

for Figure 1.

* Table 1 content are values for different parameters and not a model.

* From the 2 bottlenecks shown in Figure 2, there seems to be no difference between the 2 bottlenecks,

apart from the number of iterations (2 & 3 respectively). If that is the case, then the authors

can just use one figure and include in the explanation the 2 & 3 iterations.

* First paragraph under Discussion section, the authors should include correct measurement units for their

respective evaluation metrics.

* Correct the 1st phrase in your conclusion section. "Based on the original GhostNet,..."

* The authors should be consistent about their evaluation metrics. Under Discussion section, it was stated that

"The evaluation metrics examined in this study were FLOPs, memory usage, predicted time and Acc...".

However, under Conclusion and Abstract section, it was stated that "The performance of the Modified GhostNet

is evaluated in terms of calculation amount, memory usage, and classification speed."

Reviewer #2: The MobileNet V3 (large or small) models was not used for all possible analysis- it was selective e.g. Tables 6, 7 and 8.

I don't think your co-authors or the paper reviewers should be acknowledged at the end of the paper. Kindly remove.

In Figure 12, the third dataset is not shown.

Limit keywords to five (5).

6. PLOS authors have the option to publish the peer review history of their article (what does this mean?). If published, this will include your full peer review and any attached files.

Reviewer #1: **Yes: **Dr Olukayode Oki

Reviewer #2: No

---

## [Author Response · Author response to Decision Letter 0]

12 May 2023

Dear Editors and Reviewers:

Thank you for your letter and for the reviewers’ comments concerning our manuscript entitled “Real-time scene classification of unmanned aerial vehicles remote sensing image based on Modified GhostNet”(ID:PONE-D-23-05971). Those comments are all valuable and very helpful for revising and improving our paper as well as the important guiding significance to our researches. We have revised the title on the submission form to make it consistent with the manuscript. We have studied comments carefully and have made correction which we hope meet with approval. Revised portion are marked in yellow in the paper. The main corrections in the paper and the responds to the editors and reviewers’ comments are in the uploaded files.

---

## [Editor Report · Decision Letter 1]

25 May 2023

Real-time scene classification of unmanned aerial vehicles remote sensing image based on Modified GhostNet

PONE-D-23-05971R1

Dear Dr. Jinzhou Cao 

We’re pleased to inform you that your manuscript has been judged scientifically suitable for publication and will be formally accepted for publication once it meets all outstanding technical requirements.

Kind regards,

Sunday Adeola Ajagbe

Academic Editor

PLOS ONE
---

## [Editor Report · Acceptance letter]

29 May 2023

PONE-D-23-05971R1 

Real-time scene classification of unmanned aerial vehicles remote sensing image based on Modified GhostNet 

Dear Dr. Cao:

I'm pleased to inform you that your manuscript has been deemed suitable for publication in PLOS ONE. Congratulations! Your manuscript is now with our production department. 

Kind regards, 

on behalf of

Dr. Sunday Adeola Ajagbe 

Academic Editor

PLOS ONE